# Effects of Multi-Point Contacts during Object Contour Scanning Using a Biologically-Inspired Tactile Sensor

**DOI:** 10.3390/s20072077

**Published:** 2020-04-07

**Authors:** Lukas Merker, Sebastian J. Fischer Calderon, Moritz Scharff, Jorge H. Alencastre Miranda, Carsten Behn

**Affiliations:** 1Technical Mechanics Group, Technische Universität Ilmenau, 98693 Ilmenau, Germany; moritz.scharff@tu-ilmenau.de; 2Institute for Process Measurement and Sensor Technology, Technische Universität Ilmenau, 98693 Ilmenau, Germany; juan-sebastian.fischer-calderon@tu-ilmenau.de; 3Section of Mechanical Engineering, Pontificial Catholic University of Peru, Lima 15088, Peru; jalenca@pucp.edu.pe; 4Faculty of Mechanical Engineering, Schmalkalden University of Applied Sciences, 98574 Schmalkalden, Germany

**Keywords:** vibrissa, bio-inspired sensor, contour scanning, multi-point contact

## Abstract

Vibrissae are an important tactile sense organ of many mammals, in particular rodents like rats and mice. For instance, these animals use them in order to detect different object features, e.g., object-distances and -shapes. In engineering, vibrissae have long been established as a natural paragon for developing tactile sensors. So far, having object shape scanning and reconstruction in mind, almost all mechanical vibrissa models are restricted to contact scenarios with a single discrete contact force. Here, we deal with the effect of multi-point contacts in a specific scanning scenario, where an artificial vibrissa is swept along partly concave object contours. The vibrissa is modeled as a cylindrical, one-sided clamped Euler-Bernoulli bending rod undergoing large deflections. The elasticae and the support reactions during scanning are theoretically calculated and measured in experiments, using a spring steel wire, attached to a force/torque-sensor. The experiments validate the simulation results and show that the assumption of a quasi-static scanning displacement is a satisfying approach. Beyond single- and two-point contacts, a distinction is made between tip and tangential contacts. It is shown that, in theory, these contact phases can be identified solely based on the support reactions, what is new in literature. In this way, multipoint contacts are reliably detected and filtered in order to discard incorrectly reconstructed contact points.

## 1. Introduction

Rats and mice use their vibrissae for tactile determination of object features, e.g., shapes and textures [1,2]. In doing so, mechanical stimuli are transmitted along the hair-shaft to the follicle-sinus complex (support) of each vibrissa, where they are transduced into action potentials for further processing in the brain [3,4].

### 1.1. Vibrissa-Based Sensors

The biological principle of vibrissae frequently serves as a paragon developing tactile sensors, e.g., for object shape recognition. Even though vibrissa-inspired sensors are nowhere near from the accuracy of conventional tactile sensors, such as coordinate measuring machines, they offer a number of benefits: Firstly, the universal applicability for the detection of different object features shows an important potential. For example, the same experimental setup was used in [5] to characterize surface textures and in [6] to reconstruct object contours. Beyond that, the same setup has shown the basic suitability for flow detection in [7]. Secondly, vibrissa-sensors benefit in many ways from their highly flexible structure. Since most conventional sensors take advantage of stiff probes as force transmitters, allowing for an easy localization of objects in space, they are vulnerable to damage when accidentally colliding with an obstacle. In contrast, a flexible transmitter is advantageous in some fields of application, e.g., mobile robotics, due to its robustness against collisions. Moreover, a rigid edge probe, e.g., must be moved over an object of interest, making and releasing contact at various points in order to scan a greater part of the object. Repeatedly sensing different points does not only extend the process time, but might also require a quite sophisticated trajectory of the sensing device (co-ordinated movements of several axes). In contrast, sweeping a long and flexible rod over an object’s surface by a single continuous movement of its base relative to the object, directly provides a large number of sensed points.

Dealing with the problem of object shape scanning and reconstruction using vibrissa-inspired sensors, most publications in the field of biomimetics focus on a single artificial vibrissa. Frequently, it is considered as a continuum using linear [8,9,10,11,12] or nonlinear bending theories [5,7,13,14,15] or discretized using multi-body systems or FE-models [6,16]. In these approaches, the sophisticated kinematics of the animal’s head and vibrissae during whisking are greatly simplified, e.g., by changing either the rod’s base position or angle in order to make contact with an object. As a consequence of object contact, the rod undergoes a deformation, resulting in measurable mechanical signals at its support, which in turn can be used to draw conclusions about the contact point position. Since the force transmission from the contact point to the support is often neglected, one focus of the paper at hand is setting up a model for theoretically generating the support reactions during scanning at the base of the rod. This allows to gain a deeper understanding of the scanning process and to carry out parameter studies without the necessity of performing a wide range of experiments.

### 1.2. Multi-Point Contacts

One major limitation, which can be found in all publications mentioned above, is the restriction to single-point contact (SPC) scenarios. This means that at each time, there is only one point along the axis of the rod in contact with the object. Some authors merely hypothesize, that multi-point contacts (MPCs) at discrete points of a rod are unlikely to occur and that deflecting a rod against an object typically generates SPCs [12,16]. Other publications connect the occurrence of MPCs with the geometry of the scanned object. In [13] the authors mention, that MPCs might occur if the curvature of the scanned object approaches zero. Similarly, it is stated in [10] that the contact scenario (SPC or MPC, see ([17] Figure 9) depends on the surface slope of the scanned object. There, it was observed that SPCs primarily occurred scanning a square object, but it happened that the rod made contact with two edges of the object for special arrangements. This situation requires a closer look to refine the definition of MPC as it is understood in the present paper. The square object was scanned using a straight rod. If some point along the axis of the rod (not the tip) makes contact with the object, there may be a situation, where the straight end of the rod (between the contact position and the tip) aligns with the straight part of the object. That situation might give the impression of a line-contact or MPC, where, e.g., both edges of the square are in contact with the rod (as stated in [10]). In the presence of multiple contact forces or a distributed load, however, the straight end of the rod would also have to deform, which is not the case (see ([10] Figure 5)). Thus, there is only one actual contact point at the first position, where the rod meets the object. At this point, a single contact force acts on the rod and its straight end aligns stress-free with the object.

In [14] object contours are assumed to be strictly convex in order to exclude the occurrence of MPCs. Instead of excluding the occurrence of MPCs, some publications briefly discuss their effects. In [18] two-point contacts (TPCs) occur while the rod crosses an object-gap in context of texture determination. It is assumed, that the first contact releases abruptly when a second contact at a discrete rod position is made. This assumption may be sufficient in this particular case, where possible MPCs are very close to each other - but it is not a suitable in general. The authors in [11] relate to the principle of superposition and conclude that any load distribution along the rod can be modeled as a resultant force acting at a single point. In other words, a single contact force might cause the same mechanical signals at the base of a rod as any load distribution. This raises the question of whether it is possible to distinguish between MPCs and SPCs if the only quantities at hand are mechanical signals at the base of the rod. The authors [13,14] avoid this problem in context of object contour reconstruction by presupposing SPC scenarios. The reconstruction algorithms presented in [13,14] use the support reactions of a rod to calculate the contact force and its orientation. The clamping moment and position are used to formulate an initial-value problem, which is integrated numerically, exploiting the termination condition that the bending moment at the contact point is zero. However, this condition is only valid for SPCs. In [15] the effect of MPCs is briefly discussed using an FEA model to calculate the support reactions. Nevertheless, the investigation is limited to an example, where two discrete contact points are comparatively close to each other. Therefore, the TPC phase during scanning is limited to a negligible interval and does not allow a deeper insight into the deformation of the rod. In addition, the calculated support reactions seem to be strongly affected by the discretization of the rod (see ([15] Figure 9.22)). Finally, it is shown in ([15] Figure 9.23) that the maximum reconstruction error occurs during TPC.

For an actual sensor application, it is important that the scanned objects do not have to meet any specific requirements. If the assumption of strictly convex objectsin [15] is dropped, MPCs may occur, which influence the mechanical signals at the base of a rod in an unknown way. Using these signals to localize the contact point, the object reconstruction is affected as well, what might lead to an incorrect estimation of the scanned contour. Although, multi-point boundary-value problems (BVPs) for the rod equation were treated in some publications, e.g., [19], to date, no investigation has been carried out to clarify how MPCs affect the scanning sweep of a rod along specific objects.

### 1.3. Objective of the Present Work

The present paper contributes to the general overall goal of scanning and reconstructing arbitrary shaped object contours by means of vibrissa-inspired sensors. Here, we deal with scanning different partly concave object contours including TPC scenarios, as a first step. A single vibrissa is modeled as a rod using nonlinear bending theory. The paper particularly addresses the following goals: In Section 2.1. we derive the deformation equations of a rod with a finite number of applied forces. Secondly, the model is refined considering the rod in contact with a partly concave object contour, which is composed of two parabolas. In doing so, we identify different possible contact phases to derive the underlying BVPs. An experimental setup, consisting of a spring steel wire attached to a force/torque-sensor is used to validate the model is presented in Section 2.2. In the first part of Section 3, the deformation behavior of the rod is investigated during both contact phases, SPC and TPC. The support reactions, generated using a Matlab-algorithm, are compared with measured data. The second part of Section 3. reveals the effect of TPCs when using the reconstruction algorithm from [14]. With the goal to identify concave parts of different objects, we present an approach exploiting only the support reactions of a single rod. In doing so, we show, that it is theoretically possible, to distinguish SPC and TPC based on the support reactions exploiting a contact phase diagram. Experimental results clarify the limitations of this approach: the resolution of the used force- and torque sensor as well as frictional effects. Therefore, we suggest another way to identify concave parts of objects by repeatedly scanning at different distances.

Although the paragon in the present work is a biological one, our focus is neither on copying a vibrissa as exactly as possible nor on explaining the morphological characteristics of vibrissae. Instead, the paper aims to develop technical sensor principles in which some of the sensor properties are inspired by vibrissae. In particular, the presented approaches might be used for environmental exploration in mobile robotics. In this way, a robot equipped with vibrissa-like sensors might be adapted to deal with complex scanning situations, including large deflections in combination with TPCs.

## 2. Materials and Methods

In this section we present a mechanical model and a method of simulating scanning sweeps of an artificial vibrissa along partly convex object contours. Following, we introduce an experimental setup, which is used to validate the model as well as the simulation algorithm.

### 2.1. Modeling & Simulation

The general model, shown in Figure 1, consists of a single vibrissa modeled as an Euler-Bernoulli bending rod of length *L*, which is cylindrically shaped (causing constant characteristics of its cross section: area A=const., second moment of area $I_{z}=const.$). The rod is one-sided clamped to a horizontally movable measurement unit, see Figure 1 and consists of a homogeneous, isotropic material with a constant Young’s modulus *E*.

**Remark.** *From the outset, the following agreement is made regarding the units of measure for a dimensionless representation of the problem:*
(1)$$\begin{array}{c} \left[length\right]:=L,\;\left[forces\right]:=\frac{EI_{z}}{L^{2}},\;\left[moments\right]:=\frac{EI_{z}}{L} \end{array}$$In this way, the modeling equations within this subsection are simplified and may be used in arbitrary scalings. Afterwards, from Section 2.2 on, dimensional quantities are used in order to give the reader a more intuitive idea of the actual physical quantities. Based on this agreement, the introduction of a new notation for the dimensional quantities will be waived.

The parametrization of the rod axis by means of its slope angle φ(s), where s∈[0,1] is the natural coordinate arc length of the rod, yields: (2)$$\begin{array}{cc} \; & \frac{x(s)}{ds}=\cos\left(\varphi \left(s\right)\right) \end{array}$$(3)$$\begin{array}{cc} \; & \frac{y(s)}{ds}=\sin\left(\varphi \left(s\right)\right) \end{array}$$(4)$$\begin{array}{cc} \; & \frac{\varphi (s)}{ds}=\kappa \left(s\right) \end{array}$$

The curvature κ(s) in (4) is substituted using Euler’s constitutive law, in dimensionless representation:(5)$$\begin{array}{c} \kappa \left(s\right)=m_{z}\left(s\right) \end{array}$$
where $m_{z}\left(s\right)$ is the dimensionless bending moment of the rod’s sections. Considering the rod in Figure 1a with a finite number of *n* forces ${\vec{f}}_{j}$, acting under the angles $\alpha_{j}\in \left[-\frac{\pi }{2},\frac{\pi }{2}\right]$ at the contact positions $s_{j}\in \left(0,1\right]$, each force is represented by:$$\begin{array}{c} {\vec{f}}_{j}=f_{j}\left(\sin\left(\alpha_{j}\right){\vec{e}}_{x}-\cos\left(\alpha_{j}\right){\vec{e}}_{y}\right) \end{array}$$

Thus, the bending moment $m_{z}\left(s\right)$ for the general case of an arbitrary number of applied forces writes:(6)$$\begin{array}{c} m_{z}\left(s\right)=\left\{\begin{array}{cc} \sum_{j=1}^{n}f_{j}\left(\left(y\left(s_{j}\right)-y\left(s\right)\right)\sin\left(\alpha_{j}\right)+\left(x\left(s_{j}\right)-x\left(s\right)\right)\cos\left(\alpha_{j}\right)\right), & s\in (0,s_{1})\\ \vdots \\ \sum_{j=k}^{n}f_{j}\left(\left(y\left(s_{j}\right)-y\left(s\right)\right)\sin\left(\alpha_{j}\right)+\left(x\left(s_{j}\right)-x\left(s\right)\right)\cos\left(\alpha_{j}\right)\right), & s\in (s_{k-1},s_{k})\\ \vdots \\ f_{n}\left(\left(y\left(s_{n}\right)-y\left(s\right)\right)\sin\left(\alpha_{n}\right)+\left(x\left(s_{n}\right)-x\left(s\right)\right)\cos\left(\alpha_{n}\right)\right), & s\in (s_{n-1},s_{n})\\ 0. & s\in (s_{n},1) \end{array}\right. \end{array}$$
with 1<k<n.

In order to realize the scanning sweep, the clamping position $x_{0}$ (input variable) of the rod is shifted incrementally, translationally relative to the object contour. This process is assumed to be slow enough to treat the problem as a quasi-static one. During scanning, contacts between the rod and the object may occur at either one or even at multiple points along the rod at the same time. As a first approach within the present paper, the problem of object scanning is limited to a special object type, which is composed of two parabolas (see Figure 1b). The object is assumed to be a rigid body, whose contour g:x↦g(x) is given by a continuous, piecewise-defined function, where each sub-function applying to a certain interval of the main function is strictly convex. The arbitrary combination of two strictly convex sub-functions results in a concave area in the overall function g(x):(7)$$\begin{array}{c} g:x\mapsto g\left(x\right)=\left\{\begin{array}{cc} g_{1}\left(x\right)=5x^{2}+y_{1} & x\in \left(-\infty ,x_{int}\right)\\ g_{2}\left(x\right)=5{\left(x-x_{2}\right)}^{2}+y_{2} & x\in \left[x_{int},+\infty \right) \end{array}\right. \end{array}$$
where $\left(x_{1}=0,y_{1}\right)$ and $\left(x_{2},y_{2}\right)$ are the vertex positions of $g_{1}$ and $g_{2}$, respectively, and $x_{int}$ is the position of intersection of $g_{1}$ and $g_{2}$ (see Figure 1b). Each strictly convex sub-function is parameterized by means of its slope angle:$$\begin{array}{c} \alpha_{1}\mapsto \left(\xi_{1}\left(\alpha_{1}\right),\eta_{1}\left(\alpha_{1}\right)\right)\;\mathrm{and}\;\alpha_{2}\mapsto \left(\xi_{2}\left(\alpha_{2}\right),\eta_{2}\left(\alpha_{2}\right)\right) \end{array}$$

**Ignoring frictional effects**, the contact force is always perpendicular to the object profile tangent. The deformation of the rod results in a set of support reactions corresponding to a certain clamping position $x_{0}$. Due to the chosen object contour (Equation 7), the number of contact points is always either one or two. While TPCs always include both object parts 1 and 2 ($\alpha_{1}$ and $\alpha_{2}$), SPCs might occur either at object part 1 or 2 ($\alpha_{1}$ or $\alpha_{2}$), see Figure 2. For the sake of brevity, the index *i* is used as an abbreviation for “1or2”.

Thus, (Equation 6) simplifies to:(8)$$\begin{array}{c} m_{bz}\left(s\right)=\left\{\begin{array}{cc} f_{1}\left(\left(y\left(s_{1}\right)-y\left(s\right)\right)\sin\left(\alpha_{1}\right)+\left(x\left(s_{1}\right)-x\left(s\right)\right)\cos\left(\alpha_{1}\right)\right)\\ \;\;+f_{2}\left(\left(y\left(s_{2}\right)-y\left(s\right)\right)\sin\left(\alpha_{2}\right)+\left(x\left(s_{2}\right)-x\left(s\right)\right)\cos\left(\alpha_{2}\right)\right), & s\in (0,s_{1})\\ f_{2}\left(\left(y\left(s_{2}\right)-y\left(s\right)\right)\sin\left(\alpha_{2}\right)+\left(x\left(s_{2}\right)-x\left(s\right)\right)\cos\left(\alpha_{2}\right)\right), & s\in (s_{1},s_{2})\\ 0. & s\in (s_{2},1) \end{array}\right. \end{array}$$
for TPC, and for SPC:(9)$$\begin{array}{c} m_{bz}\left(s\right)=\left\{\begin{array}{cc} f_{i}\left(\left(y\left(s_{i}\right)-y\left(s\right)\right)\sin\left(\alpha_{i}\right)+\left(x\left(s_{i}\right)-x\left(s\right)\right)\cos\left(\alpha_{i}\right)\right), & s\in (0,s_{1})\\ 0. & s\in (s_{1},1) \end{array}\right. \end{array}$$

Substituting (Equation 8) and (Equation 9) in (Equation 5) and differentiating the curvature using (Equation 2) and (3) in order to get rid of the constants, we end up in the following differential equations describing the change of the curvature: (10)$$\begin{array}{cc} \; & \mathrm{TPC}:\;\frac{\kappa (s)}{ds}=\left\{\begin{array}{cc} f_{1}\cos\left(\varphi \left(s\right)-\alpha_{1}\right)+f_{2}\cos\left(\varphi \left(s\right)-\alpha_{2}\right), & s\in (0,s_{1})\\ f_{2}\cos\left(\varphi \left(s\right)-\alpha_{2}\right), & s\in (s_{1},s_{2})\\ 0. & s\in (s_{2},1) \end{array}\right. \end{array}$$
(11)$$\begin{array}{cc} \; & \mathrm{SPC}:\;\frac{\kappa (s)}{ds}=\left\{\begin{array}{cc} f_{i}\cos\left(\varphi \left(s\right)-\alpha_{i}\right), & s\in (0,s_{i})\\ 0. & s\in (s_{i},1) \end{array}\right. \end{array}$$

Together with (Equation 2)–(4) the deformation of the rod is represented by the following system of ordinary differential equations (ODE) of first order: (12)$$\begin{array}{cc} \; & \mathrm{TPC}:\;\left.\begin{array}{cc} x^{\prime }\left(s\right) & =\cos\left(\varphi (s)\right)\\ y^{\prime }\left(s\right) & =\sin\left(\varphi (s)\right)\\ \varphi^{\prime }\left(s\right) & =\kappa (s)\\ \kappa^{\prime }\left(s\right) & =\left\{\begin{array}{cc} f_{1}\cos\left(\varphi \left(s\right)-\alpha_{1}\right)+f_{2}\cos\left(\varphi \left(s\right)-\alpha_{2}\right), & s\in (0,s_{1})\\ f_{2}\cos\left(\varphi \left(s\right)-\alpha_{2}\right), & s\in (s_{1},s_{2})\\ 0 & s\in (s_{2},1) \end{array}\right. \end{array}\right\} \end{array}$$
and
(13)$$\begin{array}{cc} \; & \mathrm{SPC}:\;\left.\begin{array}{cc} x^{\prime }\left(s\right) & =\cos\left(\varphi (s)\right)\\ y^{\prime }\left(s\right) & =\sin\left(\varphi (s)\right)\\ \varphi^{\prime }\left(s\right) & =\kappa (s)\\ \kappa^{\prime }\left(s\right) & =\left\{\begin{array}{cc} f_{i}\cos\left(\varphi \left(s\right)-\alpha_{i}\right), & s\in (0,s_{i})\\ 0 & s\in (s_{i},1) \end{array}\right. \end{array}\right\} \end{array}$$

As Figure 2 shows, four different contact phases have to be distinguished during scanning:(a)Phase A: SPC at the tip ($s_{i}=1$) with unknown contact angle φ(1); (b)Phase B: tangential SPC at some unknown position $0<s_{i}<1$ with the contact angle condition $\varphi \left(s_{i}\right)=\alpha_{i}$; (c)Phase BA: TPC with one tangential contact at $0<s_{1}<1$ with $\varphi \left(s_{1}\right)=\alpha_{1}$ and one tip contact at $s_{2}=1$ with unknown contact angle φ(1); (d)Phase BB: TPC with two tangential contacts at $0<s_{1}<s_{2}<1$ with $\varphi \left(s_{1}\right)=\alpha_{1}$ and $\varphi \left(s_{2}\right)=\alpha_{2}$.

Note that a single scanning sweep along an object might contain various consequent contact phases. This sequence of contact phases during one scanning sweep is referred to as phase cycle in this paper.

The four contact phases in Figure 2 are characterized by the following boundary conditions (BCs):

Phase A:
(14)x(0)=x0x(1)=ξi(αi)y(0)=y0y(1)=ηi(αi)φ(0)=π/2κ(1)=0.

Phase B:
(15)x(0)=x0x(si)=ξi(αi)y(0)=y0y(si)=ηi(αi)φ(0)=π/2φ(si)=αiκ(si)=0.

Phase BA:
(16)x(0)=x0x(s1)=ξ1(α1)x(1)=ξ2(α2)y(0)=y0y(s1)=η1(α1)y(1)=η2(α2)φ(0)=π/2φ(s1)=α1κ(1)=0.

Phase BB:
(17)x(0)=x0x(s1)=ξ1(α1)x(s2)=ξ2(α2)y(0)=y0y(s1)=η1(α1)y(s2)=η2(α2)φ(0)=π/2φ(s1)=α1φ(s2)=α2κ(s2)=0.

Simulating the scanning sweep, it is not a priori known, whether the contact resulting from a certain clamping position $x_{0}$ belongs to Phase A, B, BA or BB. Thus, one major problem is to determine which of the BVPs (13)&(14), (13)&(15), (Equation 12)&(16) and (Equation 12)&(17) has to be solved. One way to address this problem is to presuppose a certain contact phase examining the solution for contradictions afterwards. Such contradictions may include the exceeding of the rod length ($s_{i}>1$) or intersections (permeation) of the calculated elastica and the object. In this way, a *Matlab*-algorithm is used to solve the unknown parameters $f_{i},\;s_{i}$ and $\alpha_{i}$ ($s_{i}$ is known in phase A) in case of SPC, and $f_{1},\;f_{2},\;s_{1},\;s_{2},\;\alpha_{1}$ and $\alpha_{2}$ ($s_{2}$ is known in phase BA) in case of TPC. The support reactions are then determined in the following way: (18)$$\begin{array}{cc} \; & \left.\begin{array}{cc} \; & f_{0x}=-f_{i}\sin\left(\alpha_{i}\right)\\ \; & f_{0y}=f_{i}\cos\left(\alpha_{i}\right)\\ \; & m_{0z}=f_{i}\;\cos\left(\alpha_{i}\right)\;\xi_{i}\left(\alpha_{i}\right)+f_{i}\;\sin\left(\alpha_{i}\right)\;\eta_{i}\left(\alpha_{i}\right) \end{array}\right\}\;\mathrm{SPC} \end{array}$$
and
(19)$$\begin{array}{cc} \; & \left.\begin{array}{cc} \; & f_{0x}=-f_{1}\sin\left(\alpha_{1}\right)-f_{2}\sin\left(\alpha_{2}\right)\\ \; & f_{0y}=f_{1}\cos\left(\alpha_{1}\right)+f_{2}\cos\left(\alpha_{2}\right)\\ \; & m_{0z}=f_{1}\;\cos\left(\alpha_{1}\right)\;\xi_{1}\left(\alpha_{1}\right)+f_{1}\;\sin\left(\alpha_{1}\right)\;\eta_{1}\left(\alpha_{1}\right)+f_{2}\;\cos\left(\alpha_{2}\right)\;\xi_{2}\left(\alpha_{2}\right)+f_{2}\;\sin\left(\alpha_{2}\right)\; \end{array}\right\}\;\mathrm{TPC} \end{array}$$

Once the support reactions are known either by simulation or measurement, they can be used to draw conclusions about the rod deformation and finally about the object’s contour by using the reconstruction algorithm presented in [14]. Since this algorithm only provides correct reconstruction results if the support reactions result from SPCs, it is necessary to distinguish SPCs and TPCs solely based on the support reactions. In [15] the following criterion for the distinction of phase A and B (SPC) was derived:(20)$$m_{0z}^{2}-2f_{0y}=0$$

It can be shown, that this criterion remains valid for the case of TPC (see Appendix A). However, the criterion (20) does not allow a distinction between SPC and TPC so far. Instead, it is used for further investigations of contact phase determination, taking advantage of the simulation results.

### 2.2. Experiments

Figure 3 shows the experimental setup that is used to validate the simulation results. A straight spring steel wire according to DIN EN 10270-1:2017-09 with a diameter of d=0.5mm and a Young’s modulus E=206GPa is used as transmitter. It is cut off from a long piece of wire and clamped by a miniature jaw chuck in a way that the free length of the wire is L=100mm. Both dimensions, diameter and length, are verified using a caliper. There is no after-treatment for the cutting edge (tip) of the wire. The small diameter of the steel wire is chosen for two reasons: firstly, with a view to the biological paragon, and secondly, because of the model assumption that object contacts are assumed to happen directly at the axis of the bending rod (geometric dimension of the diameter is neglected). The miniature jaw chuck is connected to a 3D force sensor of type K3D40 (ME-Meßsysteme), accuracy class 0.5, nominal load ±2N and a 1D torque sensor of type TD70 (ME-Meßsysteme), accuracy class 0.1, nominal load ±50Nm. The force and torque sensors are arranged in a way that the torque sensor measures signals with respect to the *z*-axis (see Figure 3). The signals are recorded using a GSV-1A4 M12/2 (ME-Meßsysteme) amplifier, a NI PXI 6221 M-Series multifunction data acquisition device and the software LabVIEW 2017 with a sampling rate of 500Hz. Due to the small diameter of the wire, a material with a large Young’s modulus is chosen in order to realize an adequate bending stiffness. In this way, the support reactions at the base of the wire are adapted for the measuring range of the available sensor equipment exploiting the well-known mechanical properties of the chosen steel wire. The entire assembly, consisting of transmitter, jaw chuck and sensors, is attached to two linear guides of type AMTEC Power Cube PLB 090 and 070 (position repeatability ±0.005mm) in serial arrangement. Three object samples with contours according to (Equation 7) are 3D-printed from Tough PLA filament using a Ultimaker S5 3D-printer. Using (Equation 7), the object contours (see Figure 3) are given by:Object 1 (O1): $\left(x_{1},y_{1}\right)=\left(0\;\mathrm{mm},0\;\mathrm{mm}\right)$, $\left(x_{2},y_{2}\right)=\left(30\;\mathrm{mm},0\;\mathrm{mm}\right)$;Object 2 (O2): $\left(x_{1},y_{1}\right)=\left(0\;\mathrm{mm},0\;\mathrm{mm}\right)$, $\left(x_{2},y_{2}\right)=\left(30\;\mathrm{mm},10\;\mathrm{mm}\right)$;Object 3 (O3): $\left(x_{1},y_{1}\right)=\left(0\;\mathrm{mm},10\;\mathrm{mm}\right)$, $\left(x_{2},y_{2}\right)=\left(30\;\mathrm{mm},0\;\mathrm{mm}\right)$.

For each experiment, the corresponding object is mounted on the platform of a hexapod of type PI M-850.50 to position and align the object relative to the linear axis of the sensor. The scanning sweeps are performed in horizontal direction (negative *x*-direction) with a constant speed of 1mm/s for three vertical object distances *q*∈[20mm,40mm,60mm] (vertical distance from the clamping to the closest parabola vertex, marked with a red point in Figure 3). The distance reference are determined by moving the undeformed wire vertically towards the objects closest vertex until the very first contact force is detected by the force sensor. Both, the objects O1, O2 and O3 and the selected object distances q were designed based on preliminary studies in order to realize well pronounced TPCs. The object distances are limited in a way that they must be greater than 20 mm to avoid plastic deformation of the wire and smaller than 60 mm to ensure the occurrence of TPCs (as a desired effect in the present paper).

Scanning sweeps are repeated three times for each of the three objects and three different object distances, starting from the very first contact between the wire and the object and terminating with a snap off of the wire from the object. This results in a total number of 27 experiments. In order to reduce dynamical effects during scanning, the experiments are performed in a way that the sensor is accelerated to the scanning velocity before the first contact with the object takes place. Due to the small scanning velocity, the impact of the first collision between the wire and the object is negligible.

## 3. Results and Discussion

The elasticae during a simulated scanning sweep (object distance q=40mm) of the rod along the objects O1, O2 and O3 are illustrated in Figure 4a–c. Note that all diagrams of Figure 4 are to be read from right to left, due to the fact that the scanning sweeps are performed in negative *x*-direction. The color of each deformation state (green, blue, orange or red) denotes the corresponding contact phase (A, B, BA or BB, respectively). Black colored deformation states represent the phase transitions. The simulation starts with the very first contact of the undeformed rod and the object and terminates at a specific position of the last equilibrium state found by the simulation algorithm.

As clarified by the Figure 4a–c, considering concave objects and large bending deflections of the rod, two-point contacts are not unlikely to occur, as presupposed in some previous publications [12,16]. Instead, the fact that two-point contacts are limited to small intervals in proportion to the whole scanning sweep indicates that three-point or even higher multi-point contacts are statistically improbable and might only occur for very special object geometries or arrangements. Figure 4d–f and Figure 4g–i show the simulated and measured support reactions corresponding to the scanning sweeps in Figure 4a–c. The simulated support reactions are represented by black lines, while the experimental data is colored in blue ($f_{0x}$), green ($f_{0y}$) and orange ($m_{0z}$). The measured data are averaged from three repeated experiments and is shown with the standard deviation (grey tube: ±3·standarddeviation). The small standard deviation and the moderate noise of the averaged data indicate a good reproducibility of the experiment. The vertical black lines represent the phase transitions corresponding to the black elasticae in Figure 4a–c. The phase transitions are determined based on the simulation. Comparing Figure 4d,g with Figure 4f,i, it can be seen that the support reactions coincide to a large extend as a consequence of the similarity between the first part of the objects O1 and O3. They start to differ with the first TPC in Figure 4a. In general, it is obvious that there is a good correlation of the simulated and experimental data. Therefore, the assumption of a quasi-static displacement seems to be a satisfying approach. Only the reaction force $f_{0y}$ shows a slight systematic deviation, with the simulated data exceeding the measured data. In order to aggregate the correlation into a single measure, we determined correlation factors using the *Matlab 2019a*-function *GoodnessOfFit* with the normalized mean square error. The correlation between simulated and experimental data of all objects and distances is shown in Figure 5.

It becomes clear that in general the measured clamping moment $m_{0z}$ correlates best with the simulated one. The reaction forces $f_{0x}$ and $f_{0y}$ also show a good correlation, except for large object distances. This is due to the fact that for larger object distances, the rod snaps from the first to the second part of the object. This dynamical transition can not be correctly described using a quasi-static model. A dynamic snap is also apparent scanning object O3: there is a small interval in Figure 4f,i, which is characterized by a large error. This deviation, even if limited to a small interval, strongly affects the correlation number in Figure 5, although there is a good overall correlation between the simulated and measured data in Figure 4f,i.

In order to demonstrate the contour reconstruction error, which arises as a consequence of TPCs, the reconstruction algorithm [14] is applied to the data in Figure 4f,i. Figure 6 shows the reconstructed contact point sequences based on the simulated data (SPC in orange and TPC in green) and experimental data (grey). The black dashed line denotes the original object contour. It becomes clear that during SPC, the object contour is well approximated. No filter or otherwise post-processing is applied to the experimental data. Even ignoring TPC, the objects O1, O2 and O3 can be clearly distinguished. Surprisingly, the dynamical snap of the rod between both parts of object O3, which caused a large error between the simulated and measured support reactions in Figure 4f,i did not cause any outliers in the reconstructed points. While the orange points coincide with the original object contour within numerical boundaries, the green contact points, based on TPC, are characterized by a large error (distance to the original contour), which is in the same order of magnitude as the experimental error. This error leads to an inaccurate estimation of the scanned object contour.

Consequently, it would be advantageous to know, which reconstructed points result from TPCs, in order to exclude them from the contour reconstruction or to modify the reconstruction algorithm for TPC (discussed in Section 4). Therefore, we discuss possibilities of contact phase determination based on the support reactions of a single rod.

In Section 2.1, a criterion for distinguishing the phases A&BA from B&BB (20) is derived. The unit related representation of (20) is either zero in case of phase B&BB or non zero in case of phase A&BA. The criterion, corresponding to the scanning sweeps in Figure 4, is plotted against the support position $x_{0}$ in Figure 7, where simulated data is colored in red and experimental data in green.

Again, the phase transitions are represented by vertical black lines. The simulated data validates (20): it is zero during phase B&BB and non zero during phase A&BA (note the small deviation from zero during the first phase A). As Figure 4b shows, the scanning sweep along object O2 does not include phase BA. Instead, TPC only occurs in phase BB. For that reason, the phase criterion deviates from 0 only in phase A. The criterion based on the experimental data deviates from zero over the entire scan. This is probably due to frictional effects which were neglected in the model. In reality, friction causes a contact force component which is tangential to the scanned object contour. Nevertheless, especially phase BA and phase A at the end of the scanning sweep can be identified by peaks in the signals of Figure 7a–c. For example, the missing peak in Figure 7b shows, that even during experiments phase C only occurred scanning object O1 and O3. In order to distinguish between SPC and TPC based on the support reactions, the contact phases are examined in more detail. While the scanning sweep along object O1 (see Figure 4a,d,g) includes all contact phases, the sweeps along the objects O2 and O3 each include only three contact phases. A more comprehensive consideration including the contact phases of all simulations can be seen in Figure 8a,b, which need to be considered in context. Figure 8a results from evaluating all simulations with regard to the occurring contact phases. Each simulated scanning sweep consists of multiple consequent contact phases. It can be categorized into one of the three listed phase cycles C1, C2 or C3 (none but these three cycles occurred). For example, the scanning sweep shown in Figure 4a passes phase cycle C2, scanning sweep Figure 4b corresponds to C1 and scanning sweep Figure 4c to C3. The assignment of all simulations is shown in dependence on the scanned object and its distance *q*. A trend can be identified which shows that for each object the phase cycle changes from C1 over C2 to C3 with an increasing object distance. The phase cycles listed in Figure 8a are combined to a single phase diagram in Figure 8b. The symbol of each phase cycle in Figure 8a indicates the path through the phase diagram in Figure 8b. As the phase diagram shows, some phase transitions can not occur. For example, if the rod is in contact phase A (SPC at the tip) at a certain clamping position $x_{0}$, it can not change directly to phase BB (tangential TPC) in the next step ($x_{0}+\Delta x_{0}$). Instead, the deformation state must change from phase A to phase B (transition A→B) before entering phase BB. The round arrows denote that the rod remains in the same contact phase as in the previous step.

Having a look at the theoretically generated support reactions in Figure 4d–i, all phase transitions (except transition A→B) caused kinks, which can be detected in the trends of the support reactions. The respective kinks can also be found in the experimental data, but can not be clearly localized due to the signal noise, occuring in every real-world measurement. However, theoretically, the final recognition of the contact phase might be made by combining the following aspects:(a)the criterion (20) for distinguishing phase A&BA from B&BB (applicable in a restricted way for the experimental data, as Figure 7 shows);(b)the phase diagram Figure 8 representing all possible phase cycles;(c)the observation that all phase transitions (except transition A→B) lead to kinks in at least one of the support reactions.

In theory, the criterion (20) can be used to determine the initial contact phase right after detecting the first contact. Within the present paper, the initial contact is always in phase A. Then, according to Figure 8b, there are only three options: the contact either remains in phase A or it changes into phase B or BA. The transition A→B might not necessarily lead to a kink in the support reactions but either way it could be detected using the criterion (20). According to the simulation results, the transition A→BA causes a kink in at least one of the support reactions, which indicates a phase transition. In this case, the criterion (20) could be used in order to distinguish phase B from BA. The idea of contact phase identification is, that in each phase, there is a maximum of two subsequent phases, which can always be distinguished using the criterion (20) (see Figure 8b). Thus, knowing the current contact phase, it is always possible to determine the subsequent phase by evaluating kinks in the support reactions and exploiting (20). In this way, our reconstruction algorithm is adapted to identify and discard the orange points in Figure 6. It can be assumed that the phase diagram in Figure 8b is not only valid if the object is composed of two parabolas, as considered here, but might also be generalized for objects composed of any two strictly convex contours. However, so far the presented procedure applies only for the idealized model. It remains to be clarified whether the kinks in the experimental support reactions Figure 4d–i can be sufficiently accurately localized using appropriate signal processing or even artificial intelligence and whether the peaks in the experimental data in Figure 7 can be exploited to distinguish between phase A&BA and B&BB. Of course, this might be facilitated by the use of sensors with enhanced resolution.

Instead, another approach including multiple scanning sweeps with different object distances might be used in order to determine the concave interval of an object contour. Figure 9a–c show the reconstructed contact points of object O1 (a), O2 (b) and O3 (c) using the support reactions from object scans with q=20mm (green), q=40mm (orange) and q=60mm (blue). In Figure 9d–f the vertical offsets between the reconstruction results were eliminated.

It appears that in general, the closest object distance *q* leads to a maximum scanning range. At the same time, a larger object distance causes the rod to penetrate deeper into the concave part of the object. This suggests that the reconstruction of concave object parts might be improved, when the rod is in tip contact (phase A or BA). Similar findings were described in [20]. As Figure 9d–f reveal the reconstructed points based on different distances are consistent to a large extend, but differ in the area of the concave contour part. Thus, a concave object part could be localized checking multiple reconstruction results for inconsistency. Once a concave interval has been localized, the incorrectly reconstructed points can be discarded here. Of course, this results in reconstruction gaps with no information about the object. However, a reconstruction gap is preferred to undetected reconstruction errors. If a reconstruction gap would occur in a real world problem, it might just be scanned more precisely, e.g., varying the object distance or the scanning direction in order to realize SPCs. Returning to the biological paragon, it is also suspected in animals, that the first scan of an object might be used for a rough localization of the object in space and a second scan to detect more detailed information [21].

Future investigations are necessary to validate the kinds of conclusions that can be drawn from this study. This includes a more detailed investigation of the rod deformation during the experiment with regard to the contact phases by means of image processing. In this way, future research should further develop and confirm the findings of the present paper, especially the phase diagram Figure 8b. One problem of identifying two-point contacts was the detection of kinks in the measured support reactions and peaks in the phase criterion. In that context, future studies should aim to investigate the potential of further signal processing or neural networks. In addition it was hypothesized in [10], that edged objects, which are not strictly convex, would produce a more sudden change of deflection, what might also result in kinks in the support reactions. Therefore, the possibility of distinguishing between edges and two-point contacts should be examined in further work.

As the results have shown, the quality of the object contour reconstruction changes with the object distance in a way that the beam better reaches concave areas of the object for larger object distances. However, this observation seems to be strongly affected by the selection of the scanned object. Therefore, the range of objects should necessarily be extended in future works. In this way it should be clarified, whether there is a generally valid limit distance, which must not be undercut, in order to realize satisfactory reconstruction results. A more advanced way to handle the reconstruction of concave objects could be to modify the reconstruction algorithm for simultaneously localizing two contact points along the axis of a rod. Preliminary investigations have shown that in case of two-point contact, there are not enough boundary conditions to reconstruct the elastica of the rod. Therefore, the mechanical model could be changed to a statically indeterminate support arrangement. This might be realized by an additional roller support at some position along the rod, where an additional reaction force could be measured and used to formulate an additional boundary condition. Either way, frictional effects should be taken into account in future works. In this way, it might be possible to improve the phase criterion, presented in this paper, for the evaluation of experimental data.

Although the focus of the present work is on developing technical sensor principles, our findings are not limited to the large scale of the steel wire we used. Instead, the dimensionless parameters introduced in Section 2 allow a scaling of the modeling equations even to the smaller dimensions of a natural vibrissa. Returning to the biological paragon, animal’s vibrissae show some geometrical features, which are neglected in the present paper. In particular, they are rather tapered and pre-curved than cylindrically shaped, as considered here [22,23]. It can be assumed, that MPCs along a tapered rod would cause more pronounced kinks, than observed in the present work. This hypothesis is based on the fact that a conical rod has a bending stiffness, which varies along its axis. Therefore, contacts at multiple discrete points along the rod would have varying degrees of influence on the support reactions. In addition, it is conceivable that certain types of pre-curvature might be suitable to allow deeper penetration of the rod into concave areas of objects. Therefore, the complexity of the model should be increased in future works taking more morphological properties of the biological paragon into account.

## 4. Conclusions

The present paper deals with the effect of multi-point contacts along the axis of a single bending rod in context of object contour scanning and reconstruction. For this purpose, the rod was one-sided clamped and swept along an object composed of two parabolas (restriction to a maximum of two contact points) translationally and quasi-statically. Deriving the modeling equations resulted in an ODE-system describing the deflection of the rod, where one equation depends on the contact scenario single- or two-point contact, respectively. The focus of investigation was on both the influence of two-point contacts on the support reactions at the clamping of the rod and on the object shape reconstruction. Beyond single- and two-point contact, further contact phases were identified: phase A—single-point contact at the tip; phase B—tangential single-point contact; phase BA—two-point contact with one contact at the tip; phase BB—two-point contact with two tangential contacts. Each contact phase is characterized by its own boundary-value problem. Simulating the scanning sweep, these four boundary-value problems were solved repeatedly using a *Matlab*-algorithm while changing the clamping position of the rod translationally. In this way, all unknown contact forces, their positions and orientations and finally the entire elastica of the rod as well as the support reactions were determined. As the results have shown, the calculated support reactions match well with the measured ones for the majority of the simulations, what validates the model and the simulation algorithm. One limitation became evident scanning objects with large distances, when dynamic transitions of the rod between both object parts occured in experiments. These dynamical effects resulted in large errors of the support reactions which, however, were limited to a small interval.

Using the support reactions in order to reconstruct the contours O1-O3, by means of an algorithm for single-point contacts, the error resulting from two-point contacts was highlighted. Since the reconstruction error might give an incorrect impression about the scanned object, we investigated, whether two-point contacts can be detected using the support reactions, in order to discard incorrectly reconstructed points. It was shown that all phase transitions, except the transition from phase A to phase B, can be identified by kinks in the simulated and experimental support reactions. In theory, a phase criterion allowed us to distinguish phase A&BA from B&BB using the simulated support reactions. In practice, frictional effects complicated the evaluation of the criterion, but, nevertheless, the presence of the phases BA and A was recognizable by peaks in the phase criterion based on the experimental data. Analyzing all simulations with regard to the occurring contact phases, only three different phase cycles (sequences of contact phases) were found. In this context, it was hypothesized that some phase transitions can not occur at all. The observed phase cycles were combined in a single phase diagram showing all possible phase transitions. Using this phase diagram, it was possible to determine the contact phase of each deformation state by evaluating kinks in the simulated support reactions and exploiting the phase criterion. Another approach of detecting concave parts of the object and thus the occurrence of two-point contacts, was realized by multiple object scans at different object distances. Finally, three different partly concave object contours were correctly reconstructed to a large extend and distinguished from each other.

## Figures and Tables

**Figure 1 sensors-20-02077-f001:**
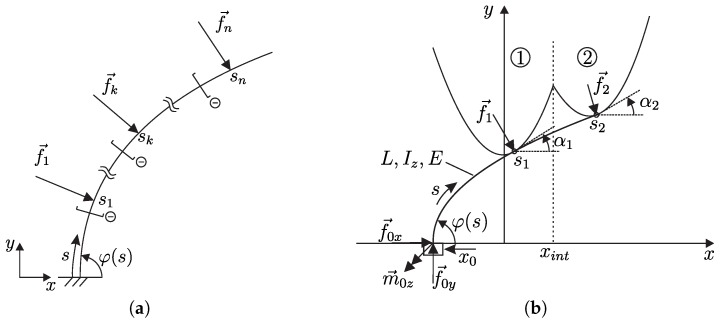
Mechanical model of a rod: (**a**) with a finite number of acting forces; (**b**) in contact with a piecewise-defined object contour function according to (Equation 7).

**Figure 2 sensors-20-02077-f002:**
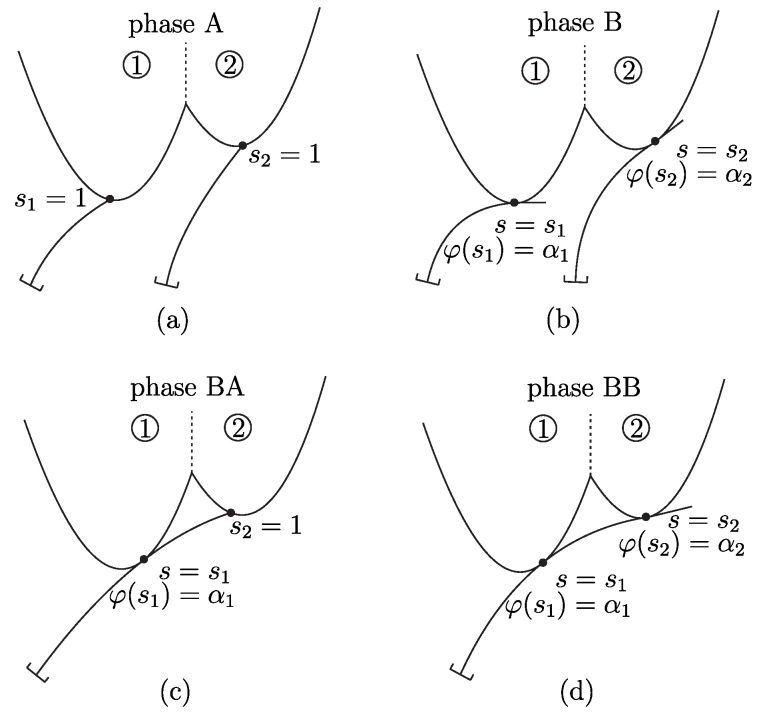
Contact phases–Phase A (**a**) and B (**b**) with single-point contact (SPC), phase BA (**c**) and BB (**d**) with two-point contact (TPC).

**Figure 3 sensors-20-02077-f003:**
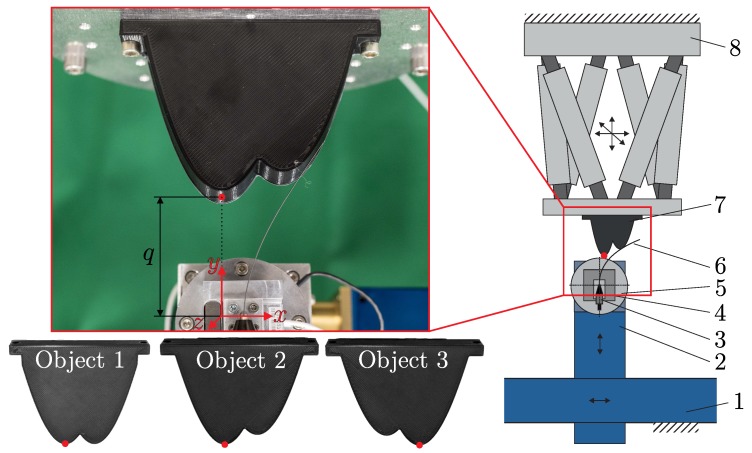
Experimental setup: 1, 2 – linear guides; 3 – jaw chuck; 4 – force sensor; 5 – torque sensor; 6 – spring steel wire; 7 – object, 8 – hexapod.

**Figure 4 sensors-20-02077-f004:**
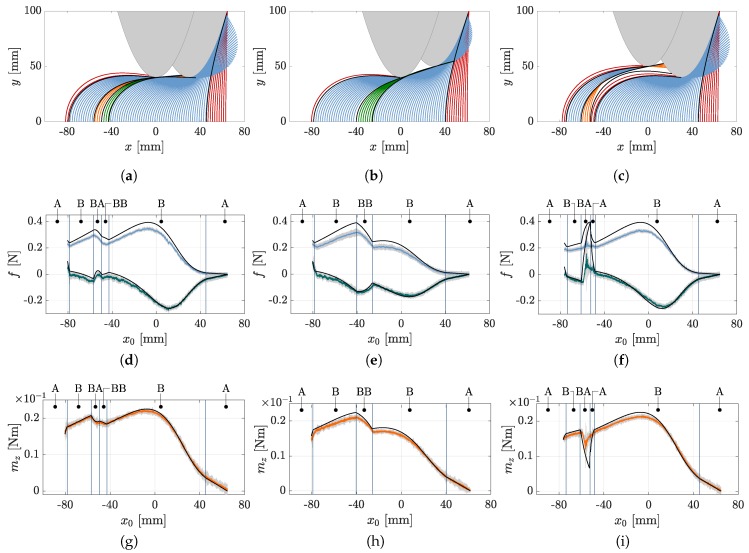
Scanning sweeps along object O1, O2 and O3: (**a**–**c**) elasticae (green: phase A, blue: phase B, orange: phase BA, red: phase BB, black: phase transitions); (**d**–**i**) simulated and measured support reactions (black: simulated data, green: measured data $f_{0x}$, blue: measured data $f_{0y}$, orange: measured data $m_{0z}$, grey: standard deviation, black vertical lines: phase transitions.

**Figure 5 sensors-20-02077-f005:**
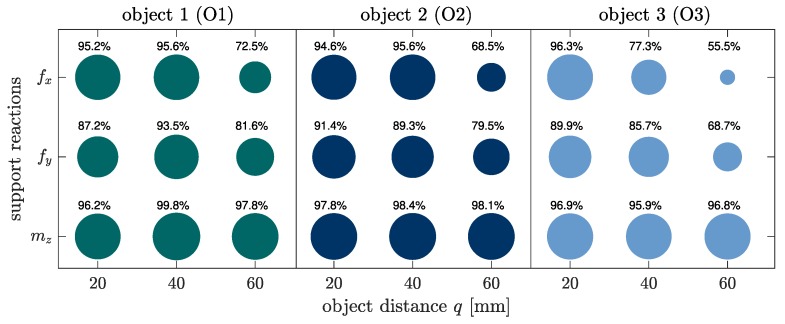
Goodness of fit between simulated and experimental data using the normalized mean square error (NMSE). The areas of the circles depend linearly on the correlation factors and are normalized to an interval [0.1,1], e.g., the smallest correlation factor corresponds to a value of 0.1, the largest one to a value of 1.

**Figure 6 sensors-20-02077-f006:**
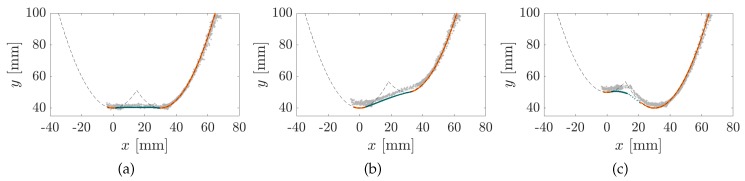
Contour reconstruction of objects O1 (**a**), O2 (**b**) and O3 (**c**) assuming SPC only: Contact points based on the simulated support reactions (SPC in orange and TPC in green) and the measured support reactions (gray).

**Figure 7 sensors-20-02077-f007:**
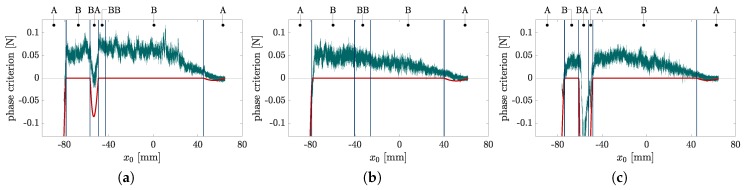
Unit related representation of phase criterion (20) for distinguishing phase A&BA from B&BB plotted against the support position $x_{0}$ (red: simulated data, green: experimental data, vertical black lines: phase transitions): scanning the objects O1 (**a**), O2 (**b**) and O3 (**c**).

**Figure 8 sensors-20-02077-f008:**
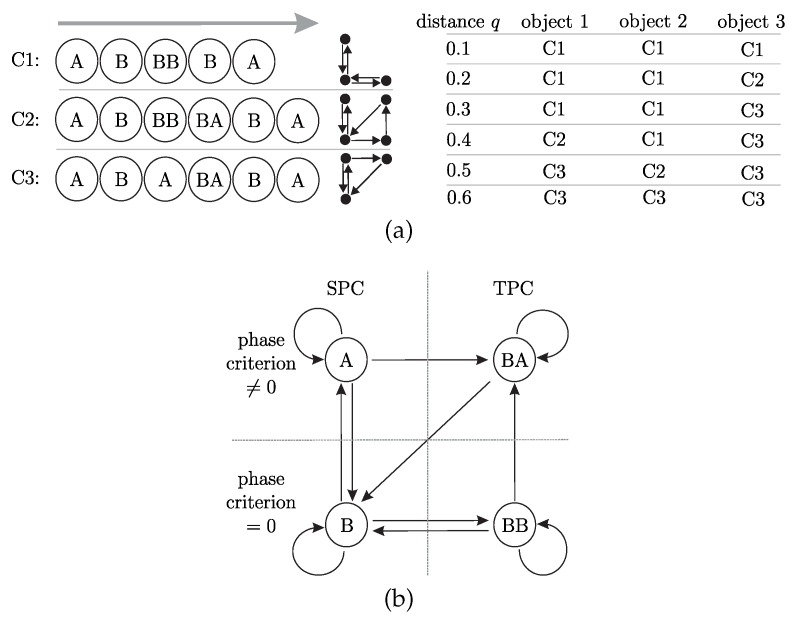
Phase transitions during object scanning: (**a**) phase cycle of each simulation in dependence on the object and its distance; (**b**) phase diagram combining all observed phase cycles.

**Figure 9 sensors-20-02077-f009:**
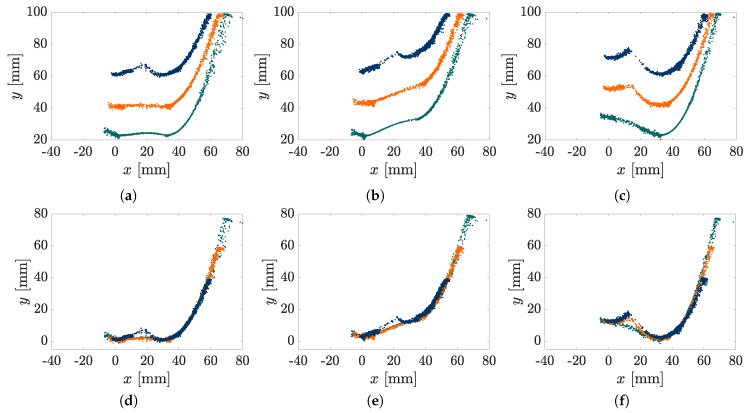
Contour reconstruction of objects O1 (**a**,**d**), O2 (**b**,**e**) and O3 (**c**,**f**) for different object distances q=20mm (green), q=40mm (orange), q=60mm(blue) (**a**–**c**) and excluding the distance offset (**d**–**f**).

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
