# Peer review of "Effects of Multi-Point Contacts during Object Contour Scanning Using a Biologically-Inspired Tactile Sensor"

_sensors, 2020, doi:10.3390/s20072077_

Round 1

Reviewer 1 Report

This paper successfully establishes analysis models and uses the modeling equations to investigate the effect of multi-point contacts along the axis of a single bending rod in the context of object contour scanning and reconstruction. It makes shape scanning and reconstruction based on vibrissa models more adaptable and accurate in dealing with complex situations. It also breaks some theoretical limitations of previous studies. I think this paper is quite interesting and meets the requirement for publication in Sensors. I recommend its acceptance subject to a minor revision, where the following aspects should be carefully addressed within a revision stage.

1. On page 3, when discussing “Objective of the present work”, some practical application scenarios and application outlooks for this method should be added and supplemented at the end.

2. On page 8, are there sufficient reasons for the selection of experimental setup materials, such as the selection of spring steel wire? Do materials with different diameters and young's modulus affect the accuracy of subsequent tests? Excluding the influence of materials and equipment settings is more helpful to illustrate the accuracy of the simulation results obtained under the model.

3. On page 9, three objects with different (x, y) settings were used for experimental verification. The set value of objects (∆x, ∆y) is the same order of magnitude as the vertical object distances q. Does it reflect that the sensor under this model is only suitable for close-range detection? Besides, the detection limit (or resolution) of the experimental setup is interesting data, so it is necessary to add some details and explanations during the experiment.

4. On page 14, the length of the conclusion seems to be a bit long, it is suggested that some experimental conclusions can be put into the corresponding experimental chapter.

Reviewer 2 Report

The authors presented an innovative design for a tactile sensor inspired by the features of vibrissae. They described its implementation, evaluated the performance of the sensor in a simulation, and validated the results in an experiment. The narrative of the paper is clear and describes the research appropriately.  

Reviewer 3 Report

Overall, the paper is well written and easy to follow. I only have a few comments for the authors’ consideration.

1.The intro provides good context for the biologically inspired approach in the paper. It would be helpful if in the Discussion or Conclusion the authors refer back to this idea of biologically inspired tactile sensing to tie it all together. It would be interesting to hear the authors’ insights on what applications they see this type of approach benefiting.

2.My interpretation of Figure 5 is that the size of the circles correspond to the goodness of fit for various conditions. It’s not clear if the circle area or the radius are being linearly scaled to the goodness of fit. I’m guessing that it’s the radius because of the drastic differences in circle size between 95.6% (fx at 40 mm) and 72.5% (fx at 60 mm). It’s a little misleading to linearly scale the radius when showing an area to represent a value. I’d suggest linearly scaling the area of the circle instead. At the very least, please be explicit about what parameter of the circle you are scaling based on the goodness of fit value.

Reviewer 4 Report

The paper presents a method to reconstruct the shape of an object when touched by a vibrissae-inspired sensor. In the specific, an in-depth study is presented to understand the effects of multiple contact points during such a task.

The paper is well written, but in some of its parts, it is not easy to follow. As an example, in the introduction, it is not so straightforward to understand the different "contact point scenarios" (in this case the reviewer suggests adding an image to better clarify the situations). Similarly, the contact phases (Line 140+) at a first read they seemed to be independent cases, instead, they are consequent events. The same happens when the Authors talk about the cycles (Lines 262+). In addition, there are some minor English errors: e.g., peaks and not peeks.

About the style, the reviewer suggests adding "Eq." or something similar to the various references in the manuscript, otherwise, they seem more references to external documents than equations.

Regarding the work itself, the reviewer has very few comments because overall is good. The reviewer's only concern is about the discrimination capabilities of the method. In fact, by considering the results, the method can not clearly distinguish between the three objects, just between two. Objects O1 and O2 are practically the same unless a geometrical transformation, i.e., a rotation. Object O3, instead, is discriminable because the seep motion is performed from right to left, and because the convex portion is larger than the other two cases. As the Authors discussed in the paper, there is a lower limit for q that enables clear discrimination between the objects. So, how about starting from a larger value of q and then try to reduce it in order to find its minimum value? By doing so, in the reviewer's opinion, will increase its impact, and improve its evaluation session.

One more question about the bio-inspiration. Natual vibrissae have a much lower Young's modulus (3-4 GPa in rats) than the one of the sensor used in this work (206 GPa); do the Author think that having a softer element will improve the resolution? 
